# Universal Health Coverage and Preparedness Payoffs: Global COVID-19 Vaccination Rates

**DOI:** 10.3390/vaccines13050443

**Published:** 2025-04-23

**Authors:** Alon Rasooly, Zachary Lebowitz, Pavel Ursu, Dorit Nitzan

**Affiliations:** 1School of Public Health, Ben Gurion University of the Negev, Beer Sheva 8410501, Israel; rasooly@post.bgu.ac.il; 2Department of Internal Medicine, Rutgers New Jersey Medical School, Newark, NJ 07103, USA; zgl4@njms.rutgers.edu; 3Division of Data, Analytics and Delivery for Impact, World Health Organization, 1211 Geneva, Switzerland; ursup@who.int

**Keywords:** universal health coverage, health system preparedness, COVID-19, pandemic response, health equity, international health regulations

## Abstract

**Objective**: The pandemic exposed global inequities in terms of health system capacities and vaccination coverage. This study evaluated the relationship between countries’ preparedness and COVID-19 vaccination rates. **Methods**: In this ecological study, we gathered global pre-pandemic data on country-level States Parties Self-Assessment Annual Reporting (SPAR) and universal health coverage (UHC) indexes. We then analyzed their relationship with COVID-19 vaccination rates in 2021–2022, using bivariate and multivariate analyses, including confounders, such as the country’s income classification and population demographics. **Findings**: The mean vaccination rates increased from 32.2% in October 2021 to 51.2% by August 2022. The UHC and SPAR indexes showed strong positive correlations with vaccination rates (r = 0.76 and r = 0.66, respectively, *p* < 0.001). In regard to the multivariate analyses, both the UHC (B = 0.81, 95% CI: 0.56–1.06) and SPAR (B = 0.34, 95% CI: 0.19–0.49) indexes remained significant predictors of vaccination rates, even after adjusting for country income level, with their influence strengthening over time, while income level effects diminished. **Conclusions**: This study underscores the critical importance of preparedness efforts, as gauged by the SPAR and UHC indexes, in shaping the effectiveness of COVID-19 vaccination responses globally. Strengthening preparedness measures is important for optimizing vaccination strategies and achieving broader immunization coverage targets.

## 1. Introduction

The COVID-19 pandemic has exposed substantial obstacles to ensuring fair and equitable vaccine access worldwide. Many nations, particularly those with fragile healthcare systems and limited resources, have grappled with the logistical complexities of procuring, distributing, and administering COVID-19 vaccines to their populations. These barriers have contributed to significant disparities in vaccination rates within individual countries and globally. The uneven vaccine rollout has exacerbated health and socioeconomic inequities among communities and countries, underscoring the urgent need for collaborative efforts to strengthen health infrastructure and address systemic challenges [1,2,3].

The critical importance of successful vaccination programs is underscored by recent evidence from the WHO European Region [4]. Researchers found that COVID-19 vaccines directly prevented approximately 1.6 million deaths between December 2020 and March 2023, reducing overall mortality by 59%. This substantial impact on survival, particularly among older adults, emphasizes the urgent need to understand factors contributing to vaccination program success across different countries.

Many low- and middle-income countries and a few high-income countries struggled to mount an effective and coordinated pandemic response due to chronic underinvestment in public health infrastructure, limited health workforce capacity, a lack of universal access to healthcare, and fragmented health governance structures [5,6]. These systemic deficiencies impeded countries’ ability to rapidly detect, prevent, and respond to the pandemic, allowing the virus to spread unchecked and overwhelm healthcare systems. It is assumed that countries with universal health coverage (UHC) are generally better positioned to bolster the overall resilience of their health systems to withstand and respond effectively to public health emergencies like the COVID-19 pandemic [5,6].

However, the WHO Director-General, Dr. Tedros Adhanom Ghebreyesus, emphasized that “Universal health coverage and health emergencies are cousins—two sides of the same coin. Strengthening health systems is the best way to safeguard against health crises [7]”. His vision has guided the current research, which aims to evaluate the relationship between countries’ COVID-19 vaccination coverage, their progress toward universal health coverage, and their preparedness for public health emergencies. Analyzing these interconnected factors provides valuable insights that can guide strategies to strengthen health systems and improve global pandemic response capabilities.

## 2. Methods

This ecological study examined relationships between health system preparedness, UHC, and COVID-19 vaccination rates across countries globally. The vaccination rates were obtained from Our World in Data [8]. Health system preparedness was measured through the use of the States Parties Self-Assessment Annual Reporting (SPAR) index from the WHO [9,10]. The SPAR index is based on countries’ self-assessments, guided by WHO-developed benchmarks to ensure consistent reporting across member states. The universal health coverage (UHC) index, which serves as the Sustainable Development Goal (SDG) indicator 3.8.1, measures average service coverage and is updated biennially according to the WHO and World Bank global monitoring reports [11]. While the WHO maintains a comprehensive COVID-19 vaccination dashboard that integrates data from multiple official sources, we utilized Our World in Data due to its more complete coverage during our study period [8,9,10,11]. Additional covariates included country income classification and population demographics from the WHO’s Global Health Expenditure Database [12] (detailed variable definitions are presented in Appendix A). Data availability varied by country and variable, resulting in different sample sizes across analyses. This variation reflects the real-world challenge in terms of data completeness in global health monitoring.

The data from different sources were combined, with countries matched by name and the International Organization for Standardization (ISO) code. The analytical approach included descriptive statistics, correlation analyses, and multiple linear regression models, which were used to examine the associations between preparedness indicators and vaccination rates. Vaccination rates (or vaccination coverage) were defined as the percentage of the population who received all doses prescribed by the initial vaccination protocol in each country, as reported by Our World in Data [8]. The models controlled for population demographics and country income classification, with analyses conducted separately for October 2021 and August 2022. In cases of missing data, the analyses included only countries with complete data on all variables of interest, with sample sizes noted for each analysis (detailed analytical procedures are provided in Appendix A). All the statistical analyses were performed using R version 4.3.2, in the RStudio environment. No ethical approval was required, since this study utilized publicly available, aggregated country-level data.

## 3. Results

Our study examined the relationship between preparedness indicators and COVID-19 vaccination rates across 183 countries. The analysis focused on two time points: October 2021 and August 2022. October 2021 marks the midpoint of global vaccination progress, when coverage rates reached half of the eventual peak levels. August 2022 represents the time when vaccination campaigns plateaued in most countries, although coverage continued to increase, particularly in low-income countries, beyond this point. The variables included the universal health coverage (UHC) index, the States Parties Self-Assessment Annual Reporting (SPAR) index, country income levels, and demographic characteristics (detailed definitions are provided in Appendix A).

The mean vaccination rates increased substantially from 32.2% in October 2021 to 51.2% by August 2022 (Table 1). The preparedness indicators showed moderate average scores, with mean SPAR and UHC indexes of 63.5 and 64.4, respectively.

The vaccination coverage by country income groups shows significant disparities (Figure 1). Upper-middle-income countries achieved vaccination rates comparable to high-income countries, while lower-middle-income countries lagged behind. Low-income countries struggled considerably, with average vaccination rates remaining below 20% by August 2022.

Preparedness indicators showed strong positive correlations with vaccination rates (Figure 2). The UHC index demonstrated robust associations (r = 0.76, *p* < 0.001 in October 2021; r = 0.73, *p* < 0.001 in August 2022). Similarly, the SPAR index showed positive correlations (r = 0.66, *p* < 0.001) at both time points (see Appendix A).

The multivariate analysis revealed that the UHC index was a significant predictor of vaccination rates in October 2021 (B = 0.81, 95% CI: 0.56–1.06, *p* < 0.001) and August 2022 (B = 1.19, 95% CI: 0.87–1.51, *p* < 0.001), even after controlling for country income level and age demographics (Table 2). The SPAR index also showed significant associations with similar statistical significance (*p* < 0.001), but lower coefficient magnitudes, at both time points (B = 0.34 and B = 0.53) compared to the UHC index (B = 0.81 and B = 1.19), indicating a smaller effect size per unit increase. The UHC and SPAR indexes showed a moderate correlation (r = 0.61), raising potential multicollinearity concerns in regard to the regression models. We therefore analyzed them using separate models (Models 1 and 2) and additionally tested their interaction using a third model (see Appendix A), which revealed no significant interaction effect, suggesting that these factors contribute independently to vaccination rates despite their correlation.

Notably, while country income level was initially a strong predictor of vaccination rates, particularly for high-income countries, its influence diminished over time. This effect was especially pronounced in models including the UHC index. The percentage of the population aged 65 and above showed no significant association with vaccination rates in our multivariate models. Our final models explained a substantial proportion of the variance in the vaccination rates, with the adjusted R-squared values ranging from 0.58 to 0.72.

## 4. Discussion

This study highlights the strong correlation between health system preparedness, UHC, and vaccination coverage, globally. High-income countries initially led the vaccination efforts, but upper-middle-income nations closed the gap over time. Low-income countries lagged behind, with vaccination rates below 20% by August 2022, underscoring global inequities.

Preparedness indicators and UHC emerged as strong predictors of vaccination coverage. The influence of national income diminished over time, suggesting that preparedness and UHC became more critical determinants of vaccination success as time went by. Despite prioritizing older adults for vaccination, the percentage of populations aged 65 and above was not significantly associated with vaccination rates, suggesting that other factors, such as health system preparedness, were more influential. The findings underscore the need for investments in UHC and preparedness to strengthen health systems for future public health emergencies.

Several studies have similarly highlighted the critical role of UHC and preparedness in improving health outcomes and pandemic responses. Lal et al. [6] emphasized that countries with stronger UHC frameworks were better equipped to manage the COVID-19 pandemic, particularly in ensuring equitable vaccine access. Their findings align with this study’s observation that higher UHC scores correlate with greater vaccination coverage, even in middle-income countries. The COVID-19 National Preparedness Collaborators [13] explored preparedness indicators, including the SPAR index, and their association with COVID-19 outcomes. Their analysis revealed that countries scoring higher in regard to preparedness metrics experienced fewer fatalities and better vaccine distribution outcomes. This aligns with the current study’s findings, which suggest that SPAR scores, although slightly less predictive than UHC, are essential for fostering coordinated health responses. The Collaborators also highlighted disparities in low-income countries, emphasizing the need for international collaboration to support under-resourced nations in building their preparedness capacity.

Similar disparities were highlighted by Santangelo et al. [14], who found that low-income countries consistently lagged behind in regard to vaccine distribution, even as global vaccination efforts progressed. Their study demonstrated that economic status initially influenced vaccination success, but that health system capacity, particularly UHC, became increasingly important over time. This shift mirrors the findings in the current study, suggesting that while the country’s economic resources provide an early advantage, long-term success in public health operations hinges also on investments in universal health coverage and health infrastructure. Both studies advocate global strategies to address inequities and strengthen preparedness in order to achieve sustainable health outcomes. Beyond direct health system capabilities, countries with established UHC systems often benefited from additional societal factors, including enhanced public trust in government health initiatives and greater social solidarity, elements that further facilitated successful vaccination campaigns [15].

Some studies have suggested different determinants of vaccination coverage. For example, Jain et al. [16] suggested that governance structures, particularly democratic governance, played a more critical role than health system indicators like UHC in determining excess mortality during the COVID-19 pandemic. They argued that political stability and transparency were better predictors of pandemic outcomes, diverging from studies prioritizing health system metrics. Similarly, Crawshaw et al. [17] focused on vaccine uptake among migrant populations in Europe and identified social determinants, such as cultural barriers, misinformation, and access to healthcare services, as more significant factors than systemic preparedness indicators like the SPAR or UHC indexes. Their work emphasized the importance of addressing societal inequities and integrating underserved populations into vaccination strategies rather than solely relying on improvements in health system preparedness. Rotaru et al. [18] also presented findings that diverge from the UHC and preparedness narrative. Their analysis of pandemic management in different countries pointed to efficient management practices and logistical coordination, rather than health system capacity, as key determinants of success. They argued that countries with well-organized pandemic response teams performed better regardless of the strengthen of the overall health system, challenging the emphasis on preparedness indexes as predictors of pandemic outcomes. These contrasting findings highlight the complexity of factors influencing health outcomes during global emergencies.

Beyond governance and structural indicators, factors such as societal trust and community engagement have emerged as critical elements in vaccination success [19]. These determinants complement health system preparedness in building resilient responses to health emergencies. The interplay between societal and structural factors is increasingly recognized in the WHO’s strategic direction, as reflected in the draft for the General Programme of Work 2025–2028 (GPW14) [20] and the WHO Results Framework [21]. Our findings on the relationship between health system preparedness and vaccination outcomes align with GPW14′s emphasis on measuring progress and strengthening data systems, providing evidence-based insights that can support countries in achieving measurable health impacts.

Our study stands out for its comprehensive approach to analyzing the role of UHC and preparedness indicators. One key factor is the period examined. We analyzed vaccination rates over an extended timeline, from October 2021 to August 2022, allowing us to observe the evolving impact of preparedness and health system capacity as global vaccination campaigns matured. Studies like the one conducted by Jain et al. [16], which emphasized the importance of governance, primarily focused on earlier stages of the pandemic, when logistical challenges and immediate response efforts may have overshadowed the influence of long-term health system preparedness.

Additionally, our study used multivariate regression models that accounted for critical confounders, including income levels, demographic factors, and interactions between the UHC and SPAR indexes (Table 2 and Appendix A). This methodological approach enabled us to isolate the specific contributions of preparedness and health system capacity to vaccination rates. In contrast, studies such as the one conducted by Crawshaw et al. [17], which focused on specific populations, like migrants, did not account for broader systemic factors that influence national-level vaccination outcomes, limiting their generalizability.

The data sources utilized further enhance the applicability of our findings. By integrating data from widely recognized repositories, including the WHO, World Bank, and “Our World in Data”, our study ensured consistency and comparability across diverse countries. This contrasts with studies like the one conducted by Rotaru et al. [18], which emphasized the importance of management practices, but relied heavily on qualitative or case-specific analyses that may not fully capture global trends. These methodological and contextual strengths make our study more relevant for policymakers seeking to design scalable, evidence-informed strategies for enhancing health system preparedness and equity in public health responses. While our study utilized Our World in Data, the WHO’s COVID-19 vaccination dashboard has emerged as a robust monitoring framework, aggregating data directly from regional offices and member states [22]. This resource could serve as a valuable tool for future studies to validate and extend our findings.

Our study recognizes its inherent limitations, but incorporates several measures to mitigate their impact and enhance the reliability of the findings. The ecological design limits the study to country-level analysis, potentially obscuring intra-country disparities. To address this, we included diverse countries across income levels and regions to capture broad global trends. Using multivariate regression models, we controlled for confounders, such as income levels, age demographics, and interactions between preparedness indicators, providing a clearer picture of the relationship between preparedness and vaccination rates. While individual- or community-level data were inaccessible, our focus on widely recognized metrics (SPAR and UHC) ensured that consistent cross-national comparisons could be made. Our ecological approach treats each country as a unit of analysis with equal weight, regardless of population size. While this is standard practice in cross-national comparative studies of health systems, it means that the findings reflect country-level patterns rather than population-weighted global trends. Future studies could explore population-weighted analyses to complement these findings. Also, the ecological design of this study cannot fully capture the unique historical, cultural, and political contexts influencing vaccine implementation in each nation. Complementary qualitative case studies would enhance the understanding of how UHC and preparedness translate into vaccination success in specific country contexts.

Publicly available data can vary in terms of quality and completeness, potentially introducing bias. To minimize this, we sourced data exclusively from established, reputable databases, including the WHO, World Bank, and “Our World in Data”, ensuring comparability and credibility. We also only included countries with complete datasets on the primary variables of interest, transparently reporting the sample sizes for each analysis to reflect variations in data availability. Recognizing the use of pre-pandemic SPAR and UHC scores, we carefully interpreted our findings within this temporal context and highlighted the need for dynamic preparedness assessments. Our study used the self-reported SPAR index rather than the newer Joint External Evaluation (JEE), which provides an independent emergency preparedness assessment. This is because pre-pandemic SPAR data were available for 172 member states compared to 106 for the JEE assessments. Future research could leverage emerging measurement frameworks being developed as part of WHO’s General Programme of Work 2025–2028 (GPW14) to assess countries’ preparedness and universal health coverage.

Although confounding factors like income level, age demographics, and population health indicators were controlled in our models, other influential variables, such as vaccine availability, geopolitical factors, and vaccine hesitancy, could not be included. To mitigate this limitation, we focused on health coverage and preparedness indicators (UHC and SPAR) that are system-level measures that are less influenced by short-term geopolitical or social fluctuations. Furthermore, our interaction terms and adjustments for income and demographic confounders helped isolate the specific effects of preparedness on vaccination outcomes.

Our analysis period (October 2021 to August 2022) allowed us to observe key trends during the global vaccine rollout, but it may not have fully captured long-term dynamics or early-stage challenges. To address this, we conducted analyses at two time points, October 2021 and August 2022, to assess how relationships between the variables evolved. This dual time-point approach provided insights into the early and later phases of vaccination campaigns, demonstrating the increasing importance of UHC and preparedness over time. The underrepresentation of some low-income countries due to incomplete data may limit the generalizability of the findings. To mitigate this, we included various countries from all income levels and transparently reported the effects of missing data on the sample size and analysis. While this limitation persists, our findings provide valuable insights for regions with complete data and emphasize the need for improved data reporting and representation from under-resourced settings. By employing rigorous statistical methods, using high-quality data sources, and transparently addressing the limitations, we sought to overcome many of the challenges inherent in ecological studies. These efforts enhance the reliability of our findings, while providing a foundation for future research to build upon, particularly in addressing granular disparities and extending analyses to dynamic pandemic contexts.

## 5. Conclusions

This study demonstrates that higher UHC and SPAR scores were strongly associated with increased vaccination coverage, particularly as the pandemic progressed. Given the demonstrated life-saving impact of COVID-19 vaccines, our findings suggest that strengthening health system preparedness and expanding universal health coverage are key to reducing vaccination inequities. This is particularly relevant for low- and middle-income countries, where enhanced health system capacity could help narrow access gaps during future health emergencies. Despite significant global progress in vaccination efforts, the disparities remain stark, with low-income countries lagging behind. These findings underscore the need for targeted international support to strengthen health systems in under-resourced regions.

To optimize global health outcomes, policymakers must prioritize the integration of UHC and act to improve national preparedness strategies. Strengthening these foundational elements will enhance the ability of health systems to deliver effective and equitable responses, ultimately contributing to broader health security and achieving sustainable development goals.

## Figures and Tables

**Figure 1 vaccines-13-00443-f001:**
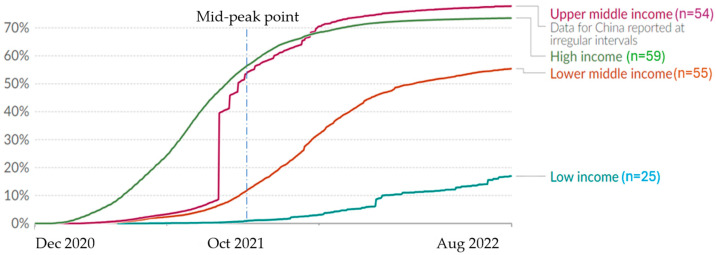
COVID-19 vaccination coverage by country income group, 2021–2022. Note: The share of the population completing the initial COVID-19 vaccination protocol according to the World Bank income group. “Mid-peak point” indicates when global coverage reached half its peak level. Source: Our World in Data.

**Figure 2 vaccines-13-00443-f002:**
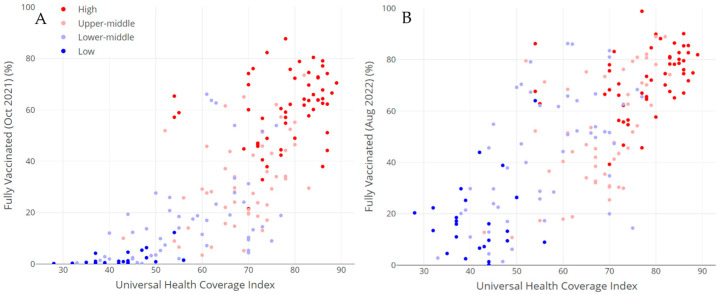
Universal health coverage and COVID-19 vaccination rates, 2021–2022. Notes: Scatter plots showing correlations between the universal health coverage index and COVID-19 vaccination rates in October 2021 (Panel (**A**), *n* = 183, r = 0.76, *p* < 0.001) and August 2022 (Panel (**B**), *n* = 178, r = 0.73, *p* < 0.001). Countries are colored according to income group. Interactive versions available as Appendix A.

**Table 1 vaccines-13-00443-t001:** Descriptive statistics for numeric variables.

Variable	Mean (SD)	Median (IQR)	*n*
Vaccination (October 2021)	32.2 (25.7)	27.7 (49.6)	183
Vaccination (August 2022)	51.2 (26.3)	53.7 (43.8)	178
SPAR	63.5 (20.3)	64.5 (32.5)	172
UHC	64.4 (15.3)	69 (24.5)	192
Above 65	8.9 (6.7)	6 (11)	190

Notes: SD = standard deviation, IQR = inter-quartile range.

**Table 2 vaccines-13-00443-t002:** Multiple linear regression results for predicting COVID-19 vaccination rates, 2021–2022 (*n* = 162).

	October 2021		August 2022	
Predictor	Model 1	Model 2	Model 1	Model 2
	B (95% CI)	B (95% CI)	B (95% CI)	B (95% CI)
UHC	0.81 ^†^ (0.56 to 1.06)		1.19 ^†^ (0.87 to 1.51)	
SPAR		0.34 ^†^ (0.19 to 0.49)		0.53 ^†^ (0.35 to 0.71)
Aged 65 and above	−0.28 (−0.74 to 0.19)	0.12 (−0.37 to 0.60)	−0.61 (−1.25 to 0.03)	0.07 (−0.59 to 0.73)
Income group (ref: low)				
Lower middle	3.22 (−4.50 to 10.94)	11.34 * (3.67 to 19.01)	9.02 (−0.39 to 18.44)	20.06 ^†^ (10.95 to 29.16)
Upper middle	8.69 (−0.51 to 17.89)	19.66 ^†^ (11.13 to 28.12)	5.80 (−5.57 to 17.18)	19.66 ^†^ (9.36 to 29.96)
High	31.53 ^†^ (20.39 to 42.67)	45.70 ^†^ (35.23 to 56.18)	18.87 * (4.96 to 32.79)	36.23 ^†^ (23.27 to 49.19)
Intercept	−31.22	−12.92	−29.94	−5.71
Adjusted R-Squared	0.71	0.69	0.58	0.56

Notes: B = unstandardized coefficient, CI = confidence interval, UHC = universal health coverage index, SPAR = States Parties Self-Assessment Annual Reporting index. Significance levels: * *p* < 0.01; ^†^ *p* < 0.001.

## Data Availability

The processed dataset used in this study is available on GitHub at https://github.com/Rasooly89/UHC-Preparedness-COVID-Vaccination-Data (accessed on 16 April 2025). The original data presented in this study are available from publicly accessible sources: COVID-19 vaccination data from Our World in Data (https://ourworldindata.org/covid-vaccinations (accessed on 28 December 2023); SPAR index data from the WHO International Health Regulations monitoring framework (https://extranet.who.int/e-spar/ (accessed on 26 February 2023); universal health coverage index from the WHO Global Health Observatory (https://www.who.int/data/gho/ (accessed on 26 February 2023); and country demographics and classifications from the WHO Global Health Expenditure Database (https://apps.who.int/nha/database (accessed on 19 February 2023).

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
