# Peer review of "Universal Health Coverage and Preparedness Payoffs: Global COVID-19 Vaccination Rates"

_vaccines, 2025, doi:10.3390/vaccines13050443_

Round 1
Reviewer 1 Report
Comments and Suggestions for Authors
This well-written and well-structured article presents an ecological study investigating the importance of pre-pandemic preparedness, as measured by Universal Health Coverage (UHC) and the State Party Annual Reporting (SPAR) indicators at the country level for COVID-19 vaccination coverage as the outcome variable in the years 2021 and 2022. It is based on data from reliable, publicly available repositories managed by the World Health Organization (WHO), the World Bank, and “Our World in Data,” encompassing practically all countries globally.
However, the manuscript has some shortcomings requiring corrections or clarifications.
Following are my comments and suggestions, from more important to minor ones:
Supplementary material includes only one figure, namely the interactive version of Panel A of Figure 2 as mentioned in the legend of Figure 2 (lines 135-136) while all the other tables and figures mentioned as included in Appendix are missing, maybe due to a technical error. Missing are: interactive Panel B of Figure 2, Figures B1-B2 mentioned in line 130, Appendix A2 (line 99) and Appendix C (line 143). Also, an Appendix containing Table A1 with detailed variable definitions is missing despite being referred to the Methods (lines 90-91) and mentioned elsewhere as Appendix A1 (line 112).
Regarding the number of countries included in different analyses:
Different numbers are mentioned: in the Highlights it reads “183 countries” (line 13) even though more countries were included (numbers varied between analyses, e.g. from 183 to 192 for different variables in Table 1. According to Table 1, vaccination coverage was known for 183 and 178 countries in October 2021 and August 2022, respectively, but at the same time, as many as 193 countries were included in Figure 1 showing cumulative vaccination coverage over time according to country income level (193=25+55+54+59, see Figure 1). (?)
Because there are altogether 195 countries in the world today including 193 United Nations member states and 2 non-member observers (the Holy See and the State of Palestine), 194 of them being members of the World Health Organization, we can say that the study encompasses practically all countries worldwide (or all WHO member states). It is reasonable that some data were missing for some countries and thus the numbers of included countries slightly differ between different analyses. However, it would be good to explain and document it better by indicating missing data and countries for each analytic procedure. Moreover, one has to bear in mind that all countries have the same weight, no matter how big or small they are (i.e. China, India, or the USA are weighted equally as Andorra, Monaco, or San Marino). Maybe that needs to be mentioned/discussed somewhere.
The following sentence might confuse the readers:
“Analysis of the potential interaction between UHC and SPAR indexes showed no statistically significant effect (see Appendix C), suggesting these factors contributed independently to vaccination rates.” (lines 142-144).
Neither the reason for such analysis of the potential interaction between UHC and SPAR nor the method or the results were mentioned in the manuscript (?). Even more surprising is why then UHC and SPAR were not included in the same multiple regression model. (?) Two different models were made and presented in Table 2, i.e. Model 1 included UHC + two other independent variables and Model 2 included SPAR + the same two independent variables. One would expect that UHC and SPAR are correlated and thus not included in the same model (?).
Regarding statistical reporting:
It is not common to report the Interquartile range (IQR) along with the mean but with the median (see Table 1 and elsewhere), and the mean usually is reported along with the standard deviation. Additionally, the abbreviation IQR is used several times before the full term (Inter-quartile range) is introduced in Table 1.
While commenting on results shown in Table 2 it is stated that “The SPAR index also showed significant, though weaker, associations at both time points …” (line 140-141) although the statistical significance level is the same, i.e. p<0.001 for both, UHC and SPAR (see Table 2). “Weaker” association would only mean that the statistical significance level is lower (e.g. p<0.01 or p<0.05), and lower unstandardized coefficients (Bs) reflect a lower expected increase of the dependent variable caused by a 1-unit increase in the corresponding independent variable (i.e. comparison between 0.34 and 0.53 for SPAR with 0.81 and 1.19 for UHC).
Multiple linear regression was employed for data analysis at two distinct time points (October 2021 and August 2022) despite the scatterplot presented in Panel A of Figure 2 showing that the association of vaccination coverage and UHC might not be linear at the first time point (October 2021) while it seems linear at the later time point (August 2022). Any comment?
Although it is well known that vaccination coverage did not change very much after December 2021, Figure 1 should be extended beyond December 2021 up to August 2022.
Regarding terminology:
Reference to the independent variable “percentage of populations aged 65 and above” is correct only once (lines 164-165) while on several other mentions of the same variable, different terms were used, e.g. “Above 65” in Table 1 and Table 2, and in line 148. Please replace “above 65” with “65 and above” everywhere. Moreover, the same variable has been differently referred to elsewhere, e.g. as “population demographics” in the Abstract (lines 29-30) and in line 96 or as “demographic characteristics” in line 111, “age demographics” in lines 140, 252, and 270, “demographic factors” in line 229, and “country demographics” in line 327. Some of these terms might be associated with an entire set of variables instead of a single variable.
“Vaccination rates” and “vaccination coverage” are referred to throughout the manuscript while the term “fully vaccinated” describes y-axis values in Figure 2. although their meaning is not clearly defined. Please introduce those terms properly (e.g. “percentage of the population completed primo-vaccination” or “percentage of people who received all doses prescribed by the initial vaccination protocol” or otherwise).
Obvious typing errors:
Line 89: references 89 and 1011
Line 310: AR: DN, and ZL instead of AR, DN, and ZL.
Author Response
Comment 1: Supplementary material includes only one figure, namely the interactive version of Panel A of Figure 2 as mentioned in the legend of Figure 2 (lines 135-136) while all the other tables and figures mentioned as included in Appendix are missing, maybe due to a technical error. Missing are: interactive Panel B of Figure 2, Figures B1-B2 mentioned in line 130, Appendix A2 (line 99) and Appendix C (line 143). Also, an Appendix containing Table A1 with detailed variable definitions is missing despite being referred to the Methods (lines 90-91) and mentioned elsewhere as Appendix A1 (line 112).
Response 1: We appreciate your thorough review that identified missing supplementary materials. We have now ensured all referenced appendices and figures are properly included in the submission. The complete set of supplementary materials now contains both interactive panels of Figure 2, Figures B1-B2, Appendices A1, A2, C, and Table A1 with variable definitions as referenced in the manuscript.
Comment 2: Regarding the number of countries included in different analyses:
Different numbers are mentioned: in the Highlights it reads “183 countries” (line 13) even though more countries were included (numbers varied between analyses, e.g. from 183 to 192 for different variables in Table 1. According to Table 1, vaccination coverage was known for 183 and 178 countries in October 2021 and August 2022, respectively, but at the same time, as many as 193 countries were included in Figure 1 showing cumulative vaccination coverage over time according to country income level (193=25+55+54+59, see Figure 1). (?)
Because there are altogether 195 countries in the world today including 193 United Nations member states and 2 non-member observers (the Holy See and the State of Palestine), 194 of them being members of the World Health Organization, we can say that the study encompasses practically all countries worldwide (or all WHO member states). It is reasonable that some data were missing for some countries and thus the numbers of included countries slightly differ between different analyses. However, it would be good to explain and document it better by indicating missing data and countries for each analytic procedure.
Response 2: Thank you for highlighting this inconsistency. We've revised the Highlights section to read 'on global scale' instead of specifying a single number. We've also added a paragraph in the Methods section explaining how data availability varied across analyses and time points. This change acknowledges the real-world challenges of data completeness in global health monitoring while providing transparency about our analytical sample sizes.
Comment 3: Moreover, one has to bear in mind that all countries have the same weight, no matter how big or small they are (i.e. China, India, or the USA are weighted equally as Andorra, Monaco, or San Marino). Maybe that needs to be mentioned/discussed somewhere.
Response 3: We appreciate this methodological insight. We have added the following sentence to the limitations paragraph in the Discussion section: 'Our ecological approach treats each country as a unit of analysis with equal weight, regardless of population size. While this is standard practice in cross-national comparative studies of health systems, it means findings reflect country-level patterns rather than population-weighted global trends. Future studies might explore population-weighted analyses to complement these findings.
Comment 4: The following sentence might confuse the readers:
“Analysis of the potential interaction between UHC and SPAR indexes showed no statistically significant effect (see Appendix C), suggesting these factors contributed independently to vaccination rates.” (lines 142-144).
Neither the reason for such analysis of the potential interaction between UHC and SPAR nor the method or the results were mentioned in the manuscript (?). Even more surprising is why then UHC and SPAR were not included in the same multiple regression model. (?) Two different models were made and presented in Table 2, i.e. Model 1 included UHC + two other independent variables and Model 2 included SPAR + the same two independent variables. One would expect that UHC and SPAR are correlated and thus not included in the same model (?).
Response 4: We appreciate this important methodological question. We've revised the Results section to clarify: 'UHC and SPAR showed moderate correlation (r=0.61), raising potential multicollinearity concerns in regression models. We therefore analyzed them in separate models (Models 1 and 2) and additionally tested their interaction in a third model (see Appendix C), which revealed no significant interaction effect, suggesting these factors contribute independently to vaccination rates despite their correlation.' This explanation has been added before discussing the regression results.
Comment 5: Regarding statistical reporting:
It is not common to report the Interquartile range (IQR) along with the mean but with the median (see Table 1 and elsewhere), and the mean usually is reported along with the standard deviation. Additionally, the abbreviation IQR is used several times before the full term (Inter-quartile range) is introduced in Table 1.
Response 5: Thank you for this statistical reporting guidance. We have revised Table 1 and other relevant sections to report median with IQR, and mean with SD where appropriate. We've also ensured the full term 'interquartile range' is defined in the notes under Table 1.
Comment 6: While commenting on results shown in Table 2 it is stated that “The SPAR index also showed significant, though weaker, associations at both time points …” (line 140-141) although the statistical significance level is the same, i.e. p<0.001 for both, UHC and SPAR (see Table 2). “Weaker” association would only mean that the statistical significance level is lower (e.g. p<0.01 or p<0.05), and lower unstandardized coefficients (Bs) reflect a lower expected increase of the dependent variable caused by a 1-unit increase in the corresponding independent variable (i.e. comparison between 0.34 and 0.53 for SPAR with 0.81 and 1.19 for UHC).
Response 6: We have revised the text to more accurately describe the results. The sentence now reads: 'The SPAR index also showed significant associations with similar statistical significance (p<0.001) but lower coefficient magnitudes at both time points (B=0.34 and B=0.53) compared to UHC (B=0.81 and B=1.19), indicating a smaller effect size per unit increase.
Comment 7: Multiple linear regression was employed for data analysis at two distinct time points (October 2021 and August 2022) despite the scatterplot presented in Panel A of Figure 2 showing that the association of vaccination coverage and UHC might not be linear at the first time point (October 2021) while it seems linear at the later time point (August 2022). Any comment?
Response 7: We note that the relationship between UHC and vaccination rates appears visually less linear at the October 2021 time point compared to August 2022, which may reflect the evolving dynamics of vaccination campaigns over time.
Comment 8: Although it is well known that vaccination coverage did not change very much after December 2021, Figure 1 should be extended beyond December 2021 up to August 2022.
Response 8: Thank you for your comment. We confirm that Figure 1 already extends through August 2022, covering the full study period as suggested.
Comment 9: Reference to the independent variable “percentage of populations aged 65 and above” is correct only once (lines 164-165) while on several other mentions of the same variable, different terms were used, e.g. “Above 65” in Table 1 and Table 2, and in line 148. Please replace “above 65” with “65 and above” everywhere. Moreover, the same variable has been differently referred to elsewhere, e.g. as “population demographics” in the Abstract (lines 29-30) and in line 96 or as “demographic characteristics” in line 111, “age demographics” in lines 140, 252, and 270, “demographic factors” in line 229, and “country demographics” in line 327. Some of these terms might be associated with an entire set of variables instead of a single variable.
Response 9: We thank the reviewer for this careful observation. We have standardized all direct references to the age variable by changing "Above 65" to "65 and above" in Table 1, Table 2, and in line 148. We have maintained broader terms such as "demographic characteristics" and "age demographics" when referring to the variable category rather than the specific measure, as these terms appropriately describe the conceptual category to which this variable belongs.
Comment 10: “Vaccination rates” and “vaccination coverage” are referred to throughout the manuscript while the term “fully vaccinated” describes y-axis values in Figure 2. although their meaning is not clearly defined. Please introduce those terms properly (e.g. “percentage of the population completed primo-vaccination” or “percentage of people who received all doses prescribed by the initial vaccination protocol” or otherwise).
Response 10: Thank you for pointing out this inconsistency. We agree that the terms "vaccination rates" and "vaccination coverage" were used interchangeably without a clear definition. We have now added a proper definition in the Methods section based on the definition already present in our Appendix A1, which clarifies the meaning of these terms throughout the manuscript.
Comment 11: Obvious typing errors:
Line 89: references 89 and 1011
Line 310: AR: DN, and ZL instead of AR, DN, and ZL.
Response 11: Thank you. We have reviewed the manuscript and corrected typing errors, including those mentioned above.
Reviewer 2 Report
Comments and Suggestions for Authors
I commend the authors for a well devised, carried out and written manuscript seeking to determine factors that predict which factors are associated with improved deployment and uptake of the COVID-19 immunizations.
They describe factors that they found to be associated with increased uptake and run a multivariate analysis to determine if they are independently associated with success.
They accessed well known databases including the OUr WOrld in Data database for vaccination rates, the WHO self-assessments for the SPAR index and the WHO Global Health Expenditure database for demographic information.
The authors show that those countries or regions with universal access to health and have health systems which have prepared for pandemic awareness as measured by the SPAR Assessment were the factors associated with increased rates.
Their multivariate analysis indicates that access to universal health and preparedness would be excellent areas for low income nations to focus based on the analysis. It was noted that High Income countries did better at the beginning of the pandemic but UHC and SPAR indexes became more important as the efforts progressed.
The authors did note that other studies have laid the focus on other areas that could impact the role out of vaccination programs. They noted that government structures, political stability, cultural barriers and the access to misinformation all have been proven to have a positive or negative effect on vaccination rates but their study was not able to incorporate these into the analysis. Additionally, those countries with efficient management and logistics capabilities show success.
Finally, they note that preparedness focusing on increasing societal trust and community engagement during pandemic preparations are factors that might overcome some of the other biases.
Congratulations to the authors this is an excellent manuscript!
Author Response
Comment 1: I commend the authors for a well devised, carried out and written manuscript seeking to determine factors that predict which factors are associated with improved deployment and uptake of the COVID-19 immunizations.
They describe factors that they found to be associated with increased uptake and run a multivariate analysis to determine if they are independently associated with success.
They accessed well known databases including the OUr WOrld in Data database for vaccination rates, the WHO self-assessments for the SPAR index and the WHO Global Health Expenditure database for demographic information.
The authors show that those countries or regions with universal access to health and have health systems which have prepared for pandemic awareness as measured by the SPAR Assessment were the factors associated with increased rates.
Their multivariate analysis indicates that access to universal health and preparedness would be excellent areas for low income nations to focus based on the analysis. It was noted that High Income countries did better at the beginning of the pandemic but UHC and SPAR indexes became more important as the efforts progressed.
The authors did note that other studies have laid the focus on other areas that could impact the role out of vaccination programs. They noted that government structures, political stability, cultural barriers and the access to misinformation all have been proven to have a positive or negative effect on vaccination rates but their study was not able to incorporate these into the analysis. Additionally, those countries with efficient management and logistics capabilities show success.
Finally, they note that preparedness focusing on increasing societal trust and community engagement during pandemic preparations are factors that might overcome some of the other biases.
Congratulations to the authors this is an excellent manuscript!
Response 1: We sincerely appreciate your positive feedback on our manuscript examining predictors of COVID-19 vaccination success. Your thoughtful assessment of our methodology and findings is greatly valued.
Reviewer 3 Report
Comments and Suggestions for Authors
Using country-level data, this study examined the associations between health system preparedness, Universal Health Coverage (UHC), and COVID-19 vaccination rates across 183 countries. Unsurprisingly, the authors found that Health system preparedness and UHC are associated with higher COVID-19 Vaccination rates. My comments are listed below
- The authors have quoted WHO director-general's words, "Universal health coverage and health emergencies are two sides of the same coin," implying the two predictors are not independent of each other. It's no surprise that both of them were associated with the outcome variable.
- Even though countries can be ranked by some metrics or indices, the correlations identified in the study are manifestations from complex mechanisms. After all, each country is unique. Sometimes, quantitative metrics and indices do not tell the whole story. Qualitative data or case studies might be able to flesh out more understandings about the variations.
- The authors did a good job in discussion of their findings. They have presented different views in explaining global variations in COVID-19 Vaccinations rates. Here I can add some of my observations from Taiwan's responses to the pandemic, where a universal health care system (the National Health Insurance) has been in place for more than 3 decades. Since the beginning of pandemic, the NHI had played an essential role in the whole preparedness system. It helped promote equitable access to care and rollout of vaccines. Because of NHI's universal coverage, it also enhanced the solidarity of the society and boosted the much-needed trust in government's handling of vaccination program. In short, there are unintended benefits embedded within a universal health system if you have one.
Author Response
Comment 1: The authors have quoted WHO director-general's words, "Universal health coverage and health emergencies are two sides of the same coin," implying the two predictors are not independent of each other. It's no surprise that both of them were associated with the outcome variable.
Response 1: Thank you for your insightful observation about the relationship between UHC and health emergencies as characterized by the WHO Director-General. While this relationship may seem intuitive, our study is the first to empirically quantify this connection on a global scale and demonstrate the specific contributions of these factors to vaccination coverage. This empirical validation provides valuable evidence-based support for policy decisions and resource allocation.
Comment 2: Even though countries can be ranked by some metrics or indices, the correlations identified in the study are manifestations from complex mechanisms. After all, each country is unique. Sometimes, quantitative metrics and indices do not tell the whole story. Qualitative data or case studies might be able to flesh out more understandings about the variations.
Response 2: Thank you for your valuable point about the complexity behind quantitative metrics. We agree that each country operates within unique contexts that standardized indices cannot fully capture. We've added an acknowledgment of this limitation and suggested that future research could benefit from qualitative case studies to provide deeper insights into country-specific factors driving vaccination success.
Comment 3: The authors did a good job in discussion of their findings. They have presented different views in explaining global variations in COVID-19 Vaccinations rates. Here I can add some of my observations from Taiwan's responses to the pandemic, where a universal health care system (the National Health Insurance) has been in place for more than 3 decades. Since the beginning of pandemic, the NHI had played an essential role in the whole preparedness system. It helped promote equitable access to care and rollout of vaccines. Because of NHI's universal coverage, it also enhanced the solidarity of the society and boosted the much-needed trust in government's handling of vaccination program. In short, there are unintended benefits embedded within a universal health system if you have one.
Response 3: Thank you for sharing your valuable observations about Taiwan's experience during the pandemic. Your insights about how established universal health coverage systems provide benefits beyond direct service delivery—such as fostering social solidarity and public trust—are particularly valuable. We've incorporated a brief statement in our discussion acknowledging these societal benefits of UHC systems that can enhance vaccination campaigns.
Reviewer 4 Report
Comments and Suggestions for Authors
I appreciate the use of publicly available data. In the interest of transparency, I would like to ask the authors to put their processed dataset and the code on github so that interested parties do not have to ask for it (I am in the process of asking other authors of other papers in the same journal and with the same data availability statement for their data and it turned out to be a hopeless task)
Author Response
Comment 1: I appreciate the use of publicly available data. In the interest of transparency, I would like to ask the authors to put their processed dataset and the code on github so that interested parties do not have to ask for it (I am in the process of asking other authors of other papers in the same journal and with the same data availability statement for their data and it turned out to be a hopeless task).
Response 1: Thank you for your suggestion regarding data transparency. We agree that making data directly accessible improves reproducibility and transparency in research. We have uploaded our processed dataset to a public GitHub repository at https://github.com/Rasooly89/UHC-Preparedness-COVID-Vaccination-Data
where it can be accessed without requiring direct contact with the authors. The repository includes documentation of all variables and original data sources.
Round 2
Reviewer 4 Report
Comments and Suggestions for Authors
The manuscript is acceptable and I appreciate the authors put the data set online.